# Persistence of the hepatic benefits of high-intensity interval training (HIIT) during detraining despite body weight regain in mice

Renata dos Santos Guarnieri⊙◉, Guilherme Sá de Oliveira⊙◉,
Kaylaine Marques Ferreira⊙, Aline Penna-de-Carvalho, Vanessa Souza-Mello,
Sandra Barbosa-da-Silva⊙*

Laboratory of Morphometry, Metabolism and Cardiovascular Disease, Institute of Biology, State University of Rio de Janeiro, Rio de Janeiro, Brazil

◉ These authors contributed equally to the preparation of this study, and both are considered first authors.
* sandra.barbosa.silva@uerj.br, sandrabarbosasilva@gmail.com

## Abstract

High-intensity interval training (HIIT) is an effective intervention for improving metabolic health and mitigating metabolic dysfunction-associated steatotic liver disease (MASLD). Nonetheless, the stability of these benefits throughout detraining periods and upon weight regain remains inadequately characterized. This study aimed to evaluate whether hepatic improvements induced by HIIT are sustained during detraining, even after body weight regain. Eighty male C57BL/6 mice were fed either a control (10% fat) or a high-fat (HF) diet (50% fat) for 12 weeks. Following this period, the animals were allocated to groups subjected to continuous HIIT or intermittent training cycles (each lasting 3 weeks). The outcomes assessed included body mass (BM), glucose tolerance, lipid profiles, liver enzyme levels (aspartate aminotransferase and alanine aminotransferase), hepatic steatosis, and the expression profiles of genes associated with lipogenesis (*Srebf1*, *Mlxpl,* and *Fas*), β-oxidation (*Ppara* and *Cpt1a*), and endoplasmic reticulum (ER) stress (*Atf4, Ddit3,* and *Gadd45*). Compared with the sedentary HF-NT condition, continuous HIIT reduced BM and improved glucose tolerance. Intermittent training (HF-TNT, HF-NTN) preserved metabolic benefits and reduced triglyceride and cholesterol levels. Notably, hepatic steatosis was significantly alleviated in all training groups but persisted even after detraining. Additionally, HIIT downregulated the expression of lipogenic genes and upregulated the expression of genes involved in β-oxidation. The levels of markers indicating ER stress were attenuated by HIIT, with a sustained reduction during periods of detraining. HIIT-induced metabolic and hepatic improvements persist partially during detraining, despite weight regain. These findings underscore the therapeutic value of continued or periodically repeated physical training in mitigating the adverse effects of an HF diet and preventing the progression of metabolic disorders such as MASLD.

**Data availability statement:** All relevant data are within the paper and its Supporting information file.

**Funding:** RSG and GSO received a bursary from the Coordenação de Aperfeiçoamento de Pessoal de Nível Superior (CAPES, Brazil), Finance Code 001. The study was supported by Fundação Carlos Chagas Filho de Amparo à Pesquisa do Estado do Rio de Janeiro (FAPERJ), grant number: 26/210.743/2024. The funders had no role in study design, analysis, decision to publish, or preparation of the manuscript.

**Competing interests:** No authors have competing interests: The authors have declared that no competing interests exist.

**Abbreviations:** AIN, American Institute of Nutrition; Acox1, Acil-CoA oxidase 1; ALT, Alanine aminotransferase; ANOVA, Analysis of variance; AST, Aspartate aminotransferase; ATF4, Activating transcription factor 4; ATP, Adenosine triphosphate; AUC, Area under the curve; BM, Body mass; BiP, Binding Immunoglobulin Protein; C, Control diet; Cd36, CD36 molecule; cDNA, Complementary DNA; Cpt1a, Carnitine palmitoyltransferase 1a; Ddit3, DNA-damage inducible transcript 3; ER, Endoplasmic reticulum; Fas, Fas cell surface death receptor; FE, Feed efficiency; FI, Food intake; Gadd45, Growth arrest and DNA-damage-inducible 45 alpha; Gapdh, Glyceraldehyde-3-phosphate dehydrogenase; IC, indirect calorimetry; HF, High-fat diet; HIIT, High-intensity interval training; HOMA-IR, Homeostasis assessment model for insulin resistance; L/BM, Liver-to-body mass; MASLD, Metabolic Dysfunction-Associated Steatotic Liver Disease; Mlxipl, MLX interacting protein-like; mRNA, Messenger RNA; NT, Untrained; OGTT, Oral glucose tolerance test; Ppara, Peroxisome proliferator-activated receptor alpha; PPAR-β/δ, Peroxisome proliferator-activated receptor-β/δ; RT-qPCR, Reverse transcriptase-quantitative PCR; RQ, respiratory quotient; Srebf1, Sterol regulatory element binding transcription factor 1; T, Trained; T2D, Type 2 diabetes; TAG, Triacylglycerol; UPR, Unfolded protein response; VO2, Maximal oxygen uptake; Vv, Volume density; WG, Weight gain.

## Introduction

Factors such as a sedentary lifestyle and unhealthy nutritional habits contribute to the increasing prevalence of overweight and obesity, which is worrisome for public health entities [1] as it creates a favorable scenario for the development and worsening of dyslipidemia and insulin resistance [2].

Among the first-line treatment alternatives for metabolic dysfunction-associated steatotic liver disease (MASLD) are lifestyle changes (such as diet and exercise) aimed at reducing body weight [3]. However, changes in daily routine can lead to interruptions in physical training. Detraining can cause the physiological and metabolic adaptations achieved to be partially or completely lost [4].

Maintaining body weight after successful weight loss requires an energy balance capable of achieving the ideal metabolism. This includes enzymatic activities and hormonal functions, synthesis and use of adenosine triphosphate (ATP), and balanced neurological and muscular functions [5].

The molecular mechanisms underlying the body's physiological efforts to regain its previous weight are not yet well understood. However, it is already known that factors such as birth type, menopause, senescence, obesogenic environmental factors, and some diseases can contribute to body weight regain [6].

Body weight regain can recur in individuals with interrupted periods of exercise, including former athletes. However, studies have indicated that the benefits of long-term training can persist during periods of physical inactivity, but there is still no consensus on exactly how long this period can last [7–10].

Detraining is associated with the progression of chronic noncommunicable diseases, such as type 2 diabetes mellitus and negatively affects insulin sensitivity and serum lipid profiles [11]. The effects of detraining appear to be directly related to the cessation time, training modality, level of physical fitness, age group, and sex [12].

Evidence suggests that previously trained skeletal muscles can regain fiber area and dynamic strength more quickly when they are subjected to exercise again after a long period of detraining [13]. This phenomenon is known as "muscle memory" and was initially conceived as the result of motor learning capacity in the central nervous system [14].

A study evaluating the effects of 4 years of uninterrupted aerobic interval training concluded that at least two training periods are necessary to efficiently improve cardiovascular risk factors. The researchers observed that in individuals who completed at least 2 cycles, consisting of 12 weeks of training, 7 weeks of detraining, and 16 weeks of retraining, their blood pressure did not return to the baseline values before the start of the training protocol. In addition, there were significant improvements in cardiorespiratory fitness, strength, biochemical markers, and body composition [15].

Although it has not yet been fully elucidated, it appears that there is a cumulative effect related to the exercise modality and the repetition of training cycles that delays the loss of the beneficial effects promoted by training. Therefore, if retraining occurs more frequently during detraining, clinical variables will return more slowly to their baseline values [16].

In light of the above, the hypothesis of the present study was that the improvements in liver parameters achieved during HIIT periods will remain, albeit partially, during the detraining periods, even if body weight is regained.

## Materials and methods

### Animal model and diets

The study was approved by the Animal Ethics Committee of the State University of Rio de Janeiro (CEUA 015/2023) and conducted in accordance with the guidelines for animal experimentation (NIH Publication No, 85−23, 1996). A total of 80 3-month-old male C57BL/6 mice were housed in microisolators on ventilated racks (Nexgen System, Allentown, Inc., PA, USA) under controlled conditions of $20 \pm 2$ °C, 60% humidity, and a light and dark cycle of 12 h, with free access to food and water.

At the beginning of the experiment (pre-HIIT phase), the mice were randomly distributed into two groups (n = 40 each) and fed different diets over the course of 12 weeks: a control diet (C, 10% of energy from lipids) or a high-fat diet (HF, 50% lipids). The diets were prepared by PragSolucoes (Jau, SP, Brazil) in accordance with the recommendations of the American Institute of Nutrition (AIN 93M) [17] (Table 1).

At the end of 12 weeks, the animals were randomly reassigned into 8 groups (n = 10) (HIIT phase), with no changes to their diets, to begin high-intensity interval training (HIIT). The groups were organized according to the application of training (T) and nontraining (N) cycles, each lasting 3 weeks (Fig 1).

Twelve-month-old C57BL/6 mice were randomly assigned to receive either a control diet (10% lipid energy) or a high-fat diet (50% lipid energy) for 12 weeks. Afterward, the mice receiving both diets were split into four groups to undergo training cycles for an additional 9 weeks.

**Table 1. Composition of Experimental Diets.**

| Ingredients (g/kg) | Diet | |
|---|---|---|
| | C | HF |
| Casein (≥ protein 85%) | 140.0 | 175.0 |
| Corn starch | 620.7 | 347.7 |
| Sucrose | 100.0 | 100.0 |
| Soybean oil | 40 | 40 |
| Lard | – | 238.0 |
| Fiber | 50.0 | 50.0 |
| Vitamin mix | 10.0 | 10.0 |
| Mineral mix | 35.0 | 35.0 |
| Cysteine | 1.8 | 1.8 |
| Choline | 2.5 | 2.5 |
| Antioxidant | 0.008 | 0.060 |
| Total (g) | 1000 | 1000 |
| Energy (kcal) | 3802.8 | 5000 |
| Carbohydrate (% energy) | 76 | 36 |
| Protein (% energy) | 14 | 14 |
| Lipid (% energy) | 10 | 50 |

Legend: Protein, mineral, and vitamin mixes were formulated according to AIN-93M. The high-fat diet (HF) contained 40% lard and 10% soybean oil, and control diet (C) contained 10% of soybean oil.

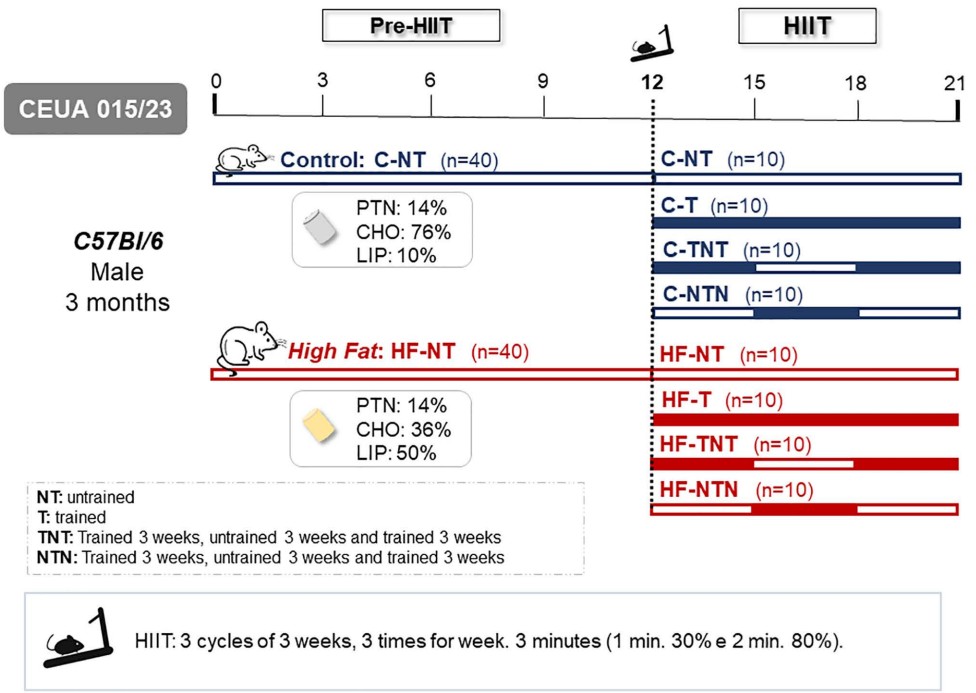

**Fig 1. Experimental design.**

## HIIT protocol

All mice assigned to the HIIT protocol were familiarized with a rodent treadmill (LE 8710–Panlab Harvard Apparatus) for five days before starting training (to 10 m/min for 10–15 min). After this familiarization, the mice were subjected to HIIT three days/week on alternating days for 12 weeks. The HIIT protocol consisted of two minutes of high intensity running at 45 m/min (90% $VO_2$) and one minute of low intensity running at 15 m/min (30% $VO_2$). The $VO_2$ peak was measured every fourth week to adjust the exercise intensity. During the HIIT period, the NT groups remained in their cages with access to water and chow ad libitum.

Intensity variations during the training sessions occurred in 3-minute cycles, with 1 minute of low intensity (30%) and 2 minutes of high intensity (80%).

## Body mass, food intake, feed efficiency, and indirect calorimetry

Body mass was measured once a week on a precision scale (B320H model, Shimadzu, Brazil), and food intake (the difference between the food offered in grams and the remaining food in the cages after 24 hours) was measured daily. In addition, feed efficiency was calculated as the ratio between weight gain and food intake and is presented as a percentage (g/kcal).

One week prior to euthanasia, respiratory metabolism was evaluated using an indirect calorimetry protocol, in which respiratory exchange was measured with the Oxylet Apparatus system (Panlab/Harvard, Barcelona, Spain), a system designed for continuous monitoring of mice with free access to food and water. Data were collected for 48 h, but data from the first 24 h were discarded because they were considered acclimation data.

## Oral glucose tolerance test

The Oral glucose tolerance test (OGTT) was performed before and after the HIIT phase. Animals were fasted for 6 hours, after which they received 2 g/kg glucose via orogastric gavage (25% in sterile saline solution, 0.9% NaCl). Blood samples

were collected via caudal puncture after 0, 15, 30, 60, and 120 minutes. Blood glucose concentrations were measured using a glucometer (Accu-Chek, Roche Diagnostics).

## Euthanasia

At the end of the experiment, after fasting for 6 hours, the mice were anesthetized with ketamine (240 mg/kg) and xylazine (30 mg/kg) via intraperitoneal injection. Blood was collected via cardiac puncture and centrifuged at room temperature to obtain plasma (712 g for 15 min), and these samples were stored individually at −80 °C until testing.

## Biochemistry

Total cholesterol, triacylglycerol, aspartate aminotransferase (AST), and alanine aminotransferase (ALT) levels were quantified using an enzymatic colorimetric method with an automatic spectrophotometer and commercial kits (Bioclin System II; Quibasa Ltda. Belo Horizonte, MG, Brazil). Leptin and insulin concentrations were quantified using commercial kits (KMP0041, Thermo Fisher Scientific;#EZRMI-13K and #EZML-82K, Millipore, MO, USA, respectively). The homeostasis assessment model for insulin resistance (HOMA-IR) index was calculated as follows: HOMA-IR index = fasting blood glucose (mmol/L) × fasting insulin concentration (µIU/L)/22.5.

## Liver

**Histopathology.** The liver was dissected and weighed. Part of each lobe was collected and fixed for 48 hours (4% w/v formaldehyde, 0.1 M phosphate buffer, pH 7.2). The remaining samples were frozen at −80 °C for molecular analysis. The fragments were embedded in Paraplast Plus (Sigma–Aldrich, St, Louis, MO, USA), sectioned at a thickness of 5 µm, and stained with hematoxylin and eosin. Digital images were captured using a BX51 microscope (Olympus, Tokyo; Japan) and an Infinity 1-5c camera (Lumenera, Ottawa, Canada).

The liver steatosis volume density (Vv [hepatic steatosis]) was assessed by counting points on a 36-point grid in at least 10 random fields per animal, as previously described [18]. Hepatic triacylglycerol concentrations were measured in liver tissue according to previously described routine protocols [19].

**Real-Time quantitative PCR (RT–qPCR).** Total RNA was extracted from the liver using TRIzol (Invitrogen, Thermo Fisher, Waltham, MA, USA). The mRNA content was measured using a NanoVue system (GE Healthcare Life Sciences, Piscataway, NJ, USA) with one microgram of RNA that had been treated with DNAse I (Invitrogen, Thermo Fisher, Waltham, MA, USA). Then, cDNA was synthesized from the sample mRNA and mixed with the primer of the gene of interest and SYBR Green mix (Invitrogen, Thermo Fisher, Waltham, MA, USA). The *Gapdh* gene was used as an endogenous control to correct the expression of the target genes. RT–qPCR was subsequently performed with a CFX96 cycler (Bio-Rad, Hercules, CA, USA). The signals were quantified using the ΔΔCT method [20]. The primers shown in Table 2 were designed using the online version of Primer3 software, version 4.0.0.

**Western blotting.** Protein levels related to lipogenesis, beta-oxidation, and endoplasmic reticulum stress were analyzed by Western blotting. Details of the primary antibodies are provided in Table 3. All target proteins were detected from the same membrane. After chemiluminescent detection of each protein, membranes were stripped using Restore™ PLUS Western Blot Stripping Buffer (MilliporeSigma), following the manufacturer's instructions, and subsequently reprobed with the corresponding primary antibodies. β-Actin was used as a loading control. Protein detection was performed by electrochemiluminescence using the ChemiDoc XRS Molecular Imaging System (Bio-Rad, Hercules, CA, USA). Band intensities (chemiluminescence signals) were quantified using ImageJ software (version 1.54g; NIH, USA).

## Data analysis

The homoscedasticity of the data distributions were tested using the Kolmogorov test. To compare data between the two groups before the start of training, a *t* test was used. After the start of training, one-way ANOVA with the Brown–Forsythe

**Table 2. Primers.**

| Genes | 5' à 3' | 3' à 5' |
|---|---|---|
| *Gapdh* | CTGACTTCAACAGCGACACC | GTGGTCCAGGGGTCTTACTC |
| *Srebf1* | AGCAGCCCCTAGAACAAACA | TCTGCCTTGATGAAGTGTGG |
| *Cherbp* | TCGAGGAAGGCACTACACCT | CACCCACTGGAAGCTGGTAT |
| *Fas* | TCGAGGAAGGCACTACACCT | CACCCACTGGAAGCTGGTAT |
| *Cd36* | GCCCAATGGAGCCATCTTTG | AGCTGCTACA GCCAGATTCA |
| *Ppara* | TCGGACTCGGTCTTCTTGAT | TCTTCCCAAAGCTCCTTCAA |
| *Cpt1a* | GCAGAGCACGGCAAAATGA | GGCTTTCGACCCGAGAAGAC |
| *Atf4* | CCGAGATGAGCTTCCTGAAC | ACCCATGAGGTTTCAAGTGC |
| *Ddit3* | CTGCCTTTCACCTTGGAGAC | CGTTTCCTGGGGATGAGATA |
| *Gadd45* | GCGAGAACGACATCAACATC | GTTCGTCACCAGCACACAGT |

Abbreviations: Glyceraldehyde-3-phosphate dehydrogenase (*Gapdh*), Sterol regulatory element-binding protein 1c (*Srebp-1c*), carbohydrate-responsive element-binding protein (*Cherbp*), fatty acid synthase (*Fas*), (*cd36*) peroxisome proliferator Activated receptors alpha (*Ppara*), carnitine palmitoyl transferase 1a (*Cpt1a*), activating transcription factor 4 (*Atf4*), *Ddit3*, DNA-damage inducible transcript 3, Growth arrest and DNA damage-inducible gene 45 (*Gadd45*).

**Table 3. Western blot antibodies.**

| Code | Antibody | Dilution | Species | Company |
|---|---|---|---|---|
| SC−81178 | β-actin | 1:500 | Anti-mouse | Santa Cruz Biotechnology, CA, USA |
| CSB-PA001172GA01HU | Acox1 | 1:700 | Anti-rabbit | Cusabio Technology, Wuhan, China |
| CSB-PA873500 | BIP/HSPA5 | 1:700 | Anti-rabbit | Cusabio Technology, Wuhan, China |
| SC-33764 | Chrebp | 1:500 | Anti-rabbit | Santa Cruz Biotechnology, CA, USA |
| SC-367 | Srebp-1 | 1:500 | Anti-rabbit | Santa Cruz Biotechnology, CA, USA |

Abbreviations: Acox, Acil-CoA oxidase 1; Srebp-1, Sterol regulatory element-binding protein 1; Chrebp,Carbohydrate-responsive element-binding protein; BIP, Binding Immunoglobulin Protein; β-actin, Beta-actin (loading control).

and Welch correction were used to make comparisons among the eight groups. During the training phase, repeated-measures one-way and two-way ANOVA followed by Tukey's post hoc test were used to analyze BM, FI, weight gain, and food efficiency (FE). Statistical analyses were performed using GraphPad Prism, version 10.2.2 (CA, USA). The results are presented as the mean ± standard deviation, and a p-value <0.05 was considered to indicate statistical significance.

## Results

### Effects of training cycles on body mass

After 12 weeks, the body mass (BM) in the HF-NT group showed a significant increase (+10%, $p = 0.003$) compared with that in the C-NT group. In contrast, continuous HIIT in the HF-T group led to a significant reduction in BM (−11%) compared with that in the HF-NT group (Fig 2), demonstrating the protective effect of uninterrupted training against excessive weight gain. The HF-TNT group exhibited a + 9% BM gain during the initial training cycle, a smaller gain of +5% during detraining (cycle 2), and a slightly greater gain (+7%) upon retraining (cycle 3), suggesting that intermittent training partially preserved metabolic benefits.

The HF-NTN group, which was only trained during cycle 2, showed the greatest BM gain (+13%) in cycle 1 (no training), a reduced gain of +4% during the training phase, and a modest increase of +5% after returning to sedentary conditions in cycle 3. The control diet groups exhibited more moderate changes in BM. The C-TNT group showed a + 5%

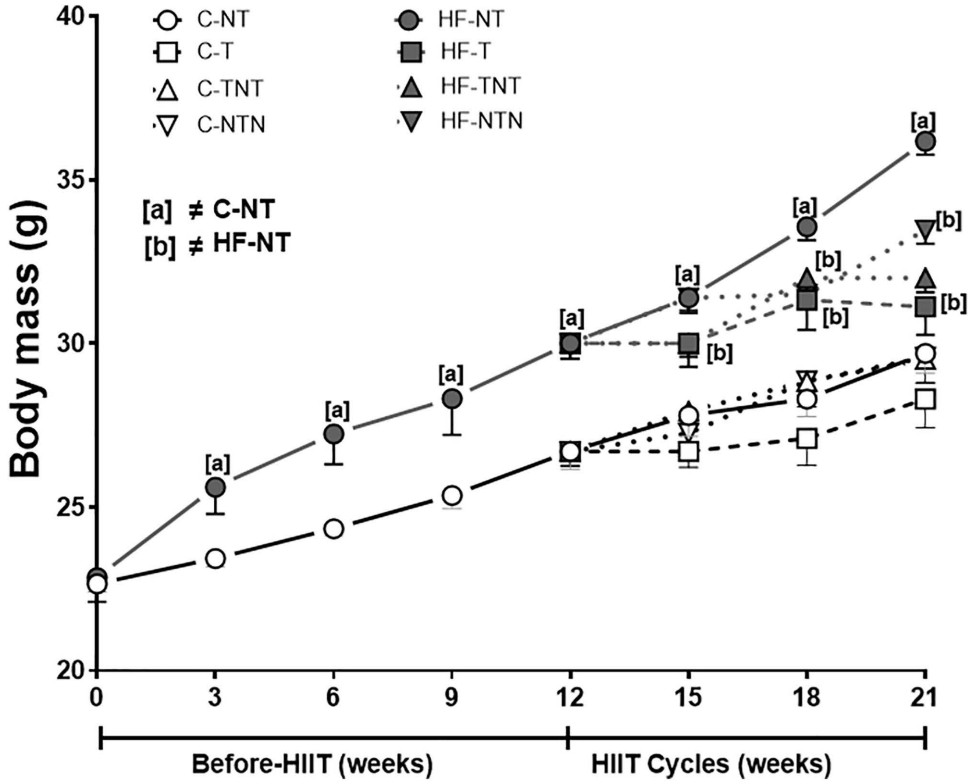

**Fig 2. Body mass evolution.** Data are presented as the mean and standard deviation, with significance determined at p<0.05; n=5 per group. The following symbols are used: [a] ≠ C-NT and [b] ≠ HF-NT. In the pre-HIIT weeks, Student's t test with Welch correction was performed for comparisons. Groups: C-NT, untrained control; C-T, control trained during the 3 cycles; C-TNT, control trained in cycle 1, not trained in cycle 2 and trained in cycle 3; C-NTN, control not trained in cycle 1, trained in cycle 2 and not trained in cycle 3; HF-NT, high-fat diet not trained; HF-T, high-fat diet trained during the 3 cycles; HF-TNT, high-fat diet trained in cycle 1, not trained in cycle 2 and trained in cycle 3; HF-NTN, high-fat diet not trained in cycle 1, trained in cycle 2 and not trained in cycle 3.

increase in cycle 1. The C-NTN group showed incremental gains of +5% (cycle 1, no training), +4% (cycle 2, training), and +3% (cycle 3, no training), indicating that the control diet mitigated excessive BM gain regardless of training status.

Two-way ANOVA revealed that both diet (*p*=0.002) and HIIT (*p*=0.003) significantly influenced BM at the end of cycle 1. However, by cycle 2, only HIIT remained significant (*p*=0.01), whereas by cycle 3, neither diet (*p*=0.99) nor HIIT (*p*=0.17) had a significant effect, suggesting a decreasing effect of interventions over time.

### Indirect calorimetry, food intake and feed efficiency

Compared with the C-NT group, the HF-NT group presented a significant reduction in the respiratory quotient (RQ; −13%, p<0.001). However, the trained groups (HF-T and HF-TNT) did not differ significantly from the HF-NT group in terms of the RQ, indicating that the substrate oxidation pattern remained similar among the HF diet groups.

With respect to energy expenditure (EE), among the HF diet-fed groups, the energy expenditure increased in the HF-T and HF-TNT groups (+11%, p=0.001 and +9%, P=0.03, respectively). In contrast, no significant difference was observed in the HF-NT group compared with the HF-NTN group (p>0.99). These findings indicate that physical training enhanced energy expenditure, an effect that persisted, although attenuated, after the detraining period. A significant reduction in food intake (g/day) was observed in the groups fed a HF diet compared with the control groups (p<0.001).

Compared with the C-NT group, the HF-NT group exhibited a significant increase in feed efficiency (+65%, p < 0.0001). Similarly, the feed efficiency in the HF-NT group was greater than that observed in the HF-T (+64%, p < 0.0001), HF-TNT (+59%, p < 0.0001), and HF-NTN (+61%, p < 0.0001) groups. In contrast, compared with the HF-TNT and HF-NTN groups, the HF-T group, subjected to continuous training, showed a marked reduction in feed efficiency (p < 0.0001), while no significant difference was observed between the HF-TNT and HF-NTN groups (p > 0.99), Table 4.

The two-way ANOVA showed that both diet and HIIT significantly affected RQ and EE (diet: p < 0.0001; HIIT: p < 0.001). Feed efficiency was also influenced by both factors (diet: p = 0.001; HIIT: p < 0.001). In contrast, food intake was primarily affected by diet (p < 0.0001), with no significant effect of HIIT (p = 0.09).

**Oral glucose tolerance test.** Before HIIT, compared with the control group, the HF group showed a 15% increase in the area under the curve (AUC). After HIIT and detraining, the AUC significantly decreased in the trained HF groups (HF-T: −31%, HF-TNT: −32%) compared with that in the untrained HF-NT group, with no significant differences among the trained HF groups. Compared with the C-NT group, the control groups (C-T and C-TNT) also showed decreases (−24% and −18%, respectively). Notably, compared with the untrained counterpart groups, the C-NTN and HF-NTN groups also showed significant AUC reductions (−16% and −30%, respectively). The HOMA-IR index, an established surrogate marker for insulin resistance, demonstrated a statistically significant reduction across all groups subjected to an HF diet and concomitant high-intensity interval training (HIIT), irrespective of whether the training was performed continuously or in cyclical patterns, Table 5. Two-way ANOVA revealed that both diet and HIIT influenced the AUC (p < 0.0001), with diet having a greater influence on cycling groups and HIIT having a greater influence on continuously trained groups (p < 0.0001).

**Lipid profile.** Compared with the C-NT group, the HF-NT group showed significantly greater plasma triacylglycerol levels (+56%). However, compared with the HF-NT group, the HF-T groups that underwent HIIT at any time point presented reductions in triacylglycerol levels: HF-T (−27%), HF-TNT (−24%), and HF-NTN (−8%). Similarly, total plasma cholesterol levels were elevated in the HF-NT group (+52%) compared with the C-NT group. HIIT reduced cholesterol levels in the HF-T (−21%) and HF-TNT (−13%) groups. Among the control diet groups, only the C-NTN group showed a modest increase (+5%, p = 0.04) compared with the C-TNT group, Table 5.

Two-way ANOVA confirmed that both diet and HIIT had significant effects on plasma triacylglycerol and total cholesterol levels (p < 0.0001 for all comparisons).

**Plasma analysis.** Compared with those in the C-NT group, the plasma insulin levels in the HF-NT group were 74% greater. In contrast, compared with HF-NT, HIIT training both continuous (HF-T) and cyclical (HF-TNT and HF-NTN) significantly reduced insulin levels, even during the detraining phase (HF-T: −22%, HF-TNT: −20%, HF-NTN: −14%). Two-way ANOVA confirmed that both diet and HIIT significantly influenced plasma insulin levels (p < 0.0001), Table 5.

**Table 4. Food intake, feed efficiency, indirect calorimetry.**

|  | C-NT | C-T | C-TNT | C-NTN | HF-NT | HF-T | HF-TNT | HF-NTN |
|---|---|---|---|---|---|---|---|---|
| Food intake (g/animal/day) | 2.6±0.01 | 2.7±0.02 | 2.7±0.02 | 2.6±0.09 | 2.5±0.05 | 2.5±0.05[a] | 2.6±0.01[a] | 2.5±0.05[a] |
| Feed efficiency (%) | 40.7±6.4 | 34.9±11.8 | 41.4±15.8 | 36.7±6.2 | 116.3±14.7 [a,e,f] | 39.1±9.6[b] | 73.0±9.6 [a,b,c] | 71.8±7.1 [a,b,c] |
| Indirect Calorimetry |  |  |  |  |  |  |  |  |
| Energy expenditure (kcal/ day/kg/^0.75) | 159.31±4.38 | 162.73±10.0 | 153.69±12.4 | 153.02±16.6 | 166.42±13.07 | 185.17±21.6 [a,b] | 182.81±8.4 [a,b] | 169.62±19.9 [a,c] |
| Respiratory quotient | 0.93±0.03 | 0.98±0.08[b] | 0.94±0.05[b] | 0.97±0.08[b] | 0.80±0.07[a] | 0.86±.04 [a] | 0.87±04 [a] | 0.85±0.05 [a] |

Legend: Data expressed as mean ± standard deviation (n = 5), with p < 0.05. Superscript letters indicate significant differences between groups: a ≠ C-NT, b ≠ HF-NT, c ≠ HF-T, d ≠ HF-TNT, e ≠ C-T, f ≠ C-TNT. Statistical analysis: one-way ANOVA with Brown-Forsythe and Welch correction.

**Table 5.  Body weight gain, glycemic profile, plasma biochemistry and hepatic parameters.**

| | C-NT | C-T | C-TNT | C-NTN | HF-NT | HF-T | HF-TNT | HF-NTN |
|---|---|---|---|---|---|---|---|---|
| **Weight gain (g)** | | | | | | | | |
| *Cycle 1* | 0.7±0.2 | 0,7±0.3 | 0.9±0.1 | 0.8±0,1 | 2.5±0.2[a] | 2.2±0.2 | 2.2±0.2 | 2.5±0.3 |
| Cycle 2 | 1.7±0.2 | 0.6±0.3[a] | 0.8±0.2[a] | 0.9±0.1[a] | 3.5±0.3[a] | 2.3±0.4[b] | 2.2±0.3[b] | 2.2±0.3[b] |
| Cycle 3 | 1.3±0.3 | −0.4±0.2[a] | −0.4±0.1[a] | 0.7±0.4[e,f] | 3.0±0.5[a] | −0.4±0.3[b] | −0.3±0.3[b] | 1.1±0.6[b,c,d] |
| **Glycemic profile** | | | | | | | | |
| Glucose (mmol/L) | 0.3±0.03 | 0.3±0.02 | 0.3±0.02 | 0.3±0.01 | 0.5±0.03[a] | 0.3±0,03[b] | 0.3±0,04[b] | 0.3±0,03[b] |
| Insulin (pg/mL) | 12.3±0.6 | 12.1±0.4 | 11.7±1.0 | 12.9±0.5 | 21.2±0.7[a] | 16.7±0.6[b] | 16.9±0.5[b] | 17.9±0.4[b] |
| HOMA-IR | 2.8±0.6 | 2.6±0.3 | 2.5±0.4 | 3.2±0.4 | 5.5±1.0[a] | 2.7±0.4[b] | 3.3±0.3[b] | 3.4±0.5[b] |
| OGTT (a.u.) | 1261±44.1 | 958±131.7 | 1014±62.2 | 1021±73.2 | 1758±187.1[a] | 1151±91.3[b] | 1119±95.5[b] | 1332±191.9[b] |
| **Plasma biochemistry** | | | | | | | | |
| Triacylglycerol (mg/dL) | 57.0±0.4 | 56.1±0.3 | 56.6±0.5 | 56.1±1.0 | 88.7±0.5[a] | 64.5±0.6[b] | 66.7±0.5[b,c] | 81.6±1.0[b,c,d] |
| Total cholesterol (mg/dL) | 114.3±5.8 | 106.3±5.1 | 110.2±2.1 | 115.3±1.4 | 173.5±5.6[a] | 136.6±3.8[b] | 150.6±4.8[b,d] | 167.2±3.8[c,d] |
| ALT (mg/dL) | 24.0±1.6 | 19.6±1.3[a] | 23.6±2.4 | 23.2±1.6 | 30.2±1.6[a] | 23.4±1.1[b] | 25.8±0.8[b] | 26.8±0.8[c] |
| AST (mg/dL) | 90.4±14.8 | 79.2±3.6 | 90.4±10.2 | 100.6±14.3 | 205.2±15.9[a] | 123.0±6.3[b] | 123.8±7.1[b] | 136.2±5.9[b] |
| Leptin (10⁻²pg/mL) | 24.1±2.7 | 15.0±0.9[a] | 17.0±0.9[a] | 24.3±2.4[d,e] | 149.2±5.8[a] | 54.5±7.9[b] | 58.6±11.3[b] | 81.5±6.2[b,c] |
| **Hepatic parameters** | | | | | | | | |
| Ratio liver/body mass (%) | 0.43±0.004 | 0.42±0.007 | 0.43±0.005 | 0.43±0.005 | 0.62±0.02[a] | 0.47±0.008[b] | 0.52±0.01[b,c] | 0.54±0.007[b,c,d] |
| Hepatic triacylglycerol (mg/g) | 1.47±0.07 | 1.39±0.09 | 1.47±0.05 | 1.47±0.06 | 2.43±0.09[a] | 1.81±0.03[b] | 1.88±0.1[b,d] | 2.1±0.07[b,c,d] |
| Hepatic Steatosis (%) | 13.7±1.05 | 10.54±1.08 [a] | 11.68±1.02 | 13.1±0.44 | 44.2±4.15[a] | 30.3±1.19[b] | 30.44±1.04[b] | 36.44±3.57[a] |

Legend: Data are expressed as mean±standard deviation (n=5). Statistical significance was set at p<0.05. Different superscript letters indicate significant differences between groups: a≠C-NT, b≠HF-NT, c≠HF-T, d≠HF-TNT, e≠C-T, f≠C-TNT. Statistical analysis used: one-way ANOVA and Brown-Forsythe and Welch correction.

In terms of the hepatic enzymes AST and ALT, compared with the C-NT group, the HF-NT group presented significantly elevated liver enzyme levels, with increases in the AST and ALT levels of 126% and 26%, respectively. In contrast, compared with HF-NT, HIIT training, both continuously (HF-T) and cyclically (HF-TNT and HF-NTN), led to marked reductions in the AST (−41%, −39%, and −34%, respectively) and ALT (−22%, −15%, and −11%, respectively) levels.

In the control diet groups, compared with the sedentary counterpart group (C-NT), animals that underwent HIIT (C-T) also presented decreases in the AST (−18%) and ALT (−10%) levels. Two-way ANOVA confirmed that both diet and HIIT significantly influenced AST and ALT levels (p<0.0001 for both factors), indicating that exercise and dietary composition play key roles in modulating liver health.

Compared with the C-NT group, the HF-NT group exhibited a markedly greater leptin concentration, with an increase of 520%. However, all HF diet-fed groups that underwent HIIT, whether continuously (HF-T) or in cycles (HF-TNT and HF-NTN), showed significant reductions in leptin levels relative to those in the HF-NT group. Specifically, leptin decreased by 63% in the HF-T group, 60% in the HF-TNT group, and 45% in the HF-NTN group, Table 5.

Two-way ANOVA confirmed that both diet (p<0.0001) and HIIT (p<0.0001) significantly influenced the leptin concentration.

**Liver parameters.**  Compared with the C-NT group, the HF-NT group exhibited a significant increase (+45%) in the liver-to-body mass (L/BM) ratio, Table 5. This ratio was lower in the HF-T (−21%), HF-TNT (−16%), and HF-NTN (−13%) groups than in the HF-NT group. It is important to note that the L/BM ratio refers to the liver weight normalized to the total body weight, not absolute liver mass.

Similarly, the liver triacylglycerol content was markedly elevated (+64%) in the HF-NT group compared with the C-NT group, Table 5. HIIT training led to notable reductions in hepatic triacylglycerol levels −25% in the HF-T group, −22% in the HF-TNT group, and −13% in the HF-NTN group when compared with those in the HF-NT group.

Two-way ANOVA revealed that both diet ($p < 0.0001$) and HIIT ($p < 0.0001$) significantly influenced the L/BM ratio and hepatic triacylglycerol content (diet: $p < 0.0001$; HIIT: $p < 0.0001$).

Compared with the C-NT group, the HF-NT group presented a markedly greater hepatic steatosis density, with an increase of +221% (Table 5, Fig 3). HIIT significantly attenuated steatosis in the HF-T group, with a reduction of −31% relative to that in the HF-NT group. Notably, both groups that underwent training cycles interspersed with detraining also exhibited improvements: compared with the HF-NT group, the HF-TNT group showed a −31% reduction, and the HF-NTN group showed a −17% decrease. Interestingly, there was no significant difference in hepatic steatosis between the HF-T and HF-TNT groups ($p = 0.45$). This finding suggests that two cycles of HIIT were as effective as continuous training in reducing steatosis, despite the presence of a detraining phase. Two-way ANOVA confirmed that both diet ($p < 0.0001$) and HIIT ($p < 0.0001$) significantly affected hepatic steatosis levels.

**Hepatic lipogenic pathway (RT-qPCR).** Compared with the C-NT group, the HF-NT group exhibited significantly upregulated expression of lipogenic and lipid transport genes, i.e., *Srebf1* (+38%), *Mlxpl* (+146%), *Fas* (+134%), and *Cd36* (+145%) (Fig 4). In contrast, continuous HIIT in the HF-T group led to marked reductions in the expression of these genes, i.e., −25% (*Srebf1*), −31% (*Mlxpl*), −28% (*Fas*), and −32% (*Cd36*), relative to those in the HF-NT group. Interestingly, compared with the HF-NT group, which underwent alternating cycles of training, detraining, and retraining, the HF-T group also showed a downregulation in gene expression levels similar to those in the HF-T group, i.e., −19% (*Srebf1*), −31% (*Mlxpl*), −27% (*Fas*), and −32% (*Cd36*), respectively, compared with those in the HF-NT group. This suggests that partial training preserved the beneficial transcriptional effects of continuous HIIT.

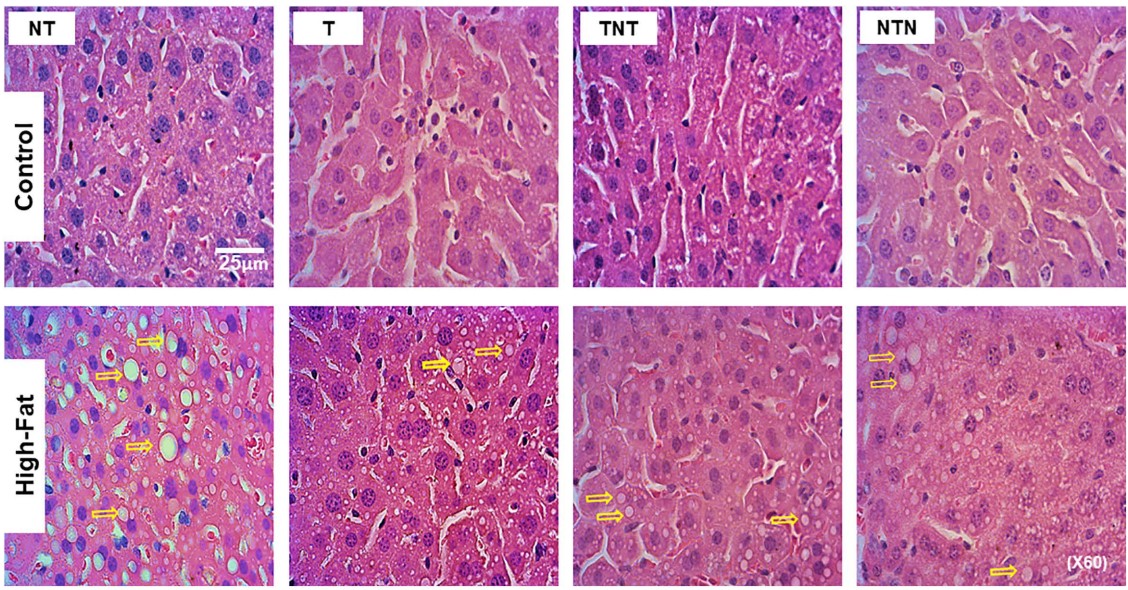

**Fig 3. Hepatic steatosis.** Photomicrographs of liver tissue stained with H&E showing microdroplets (indicated by arrows) and some larger lipid macrodroplets (arrowheads) within hepatocytes in the HF-NT group. In contrast, the liver parenchyma was well preserved in the C-NT group. Notably, compared with the HF-NT group, the HF-T group presented a reduced presence of lipid droplets. All images were captured at the same magnification (60x), and the scale bar represents 25 μm.

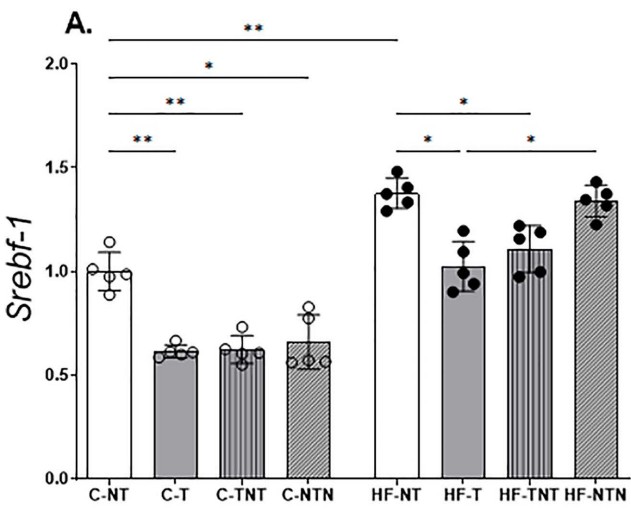

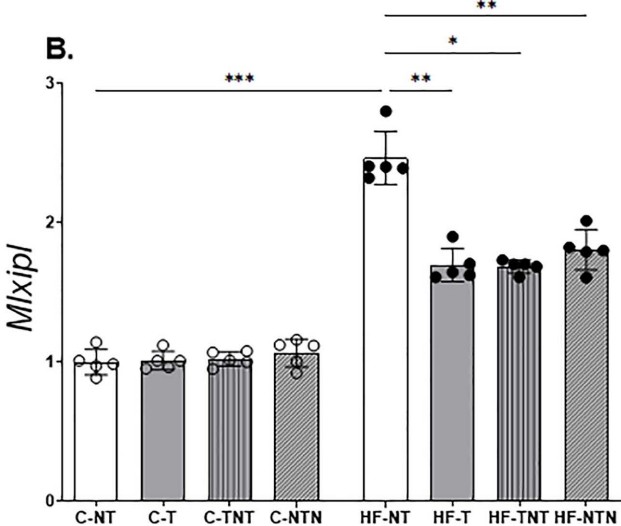

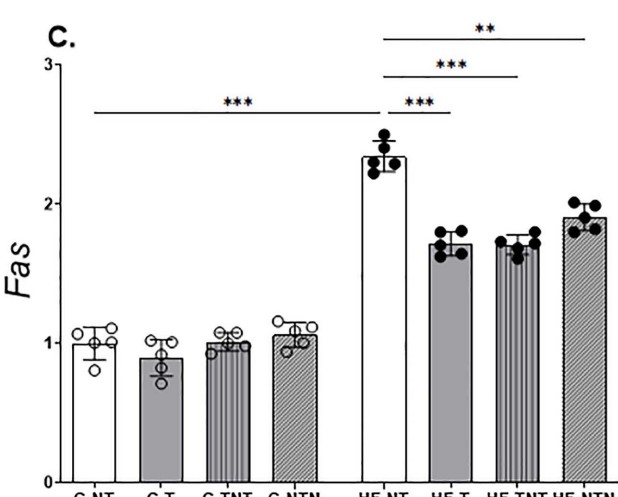

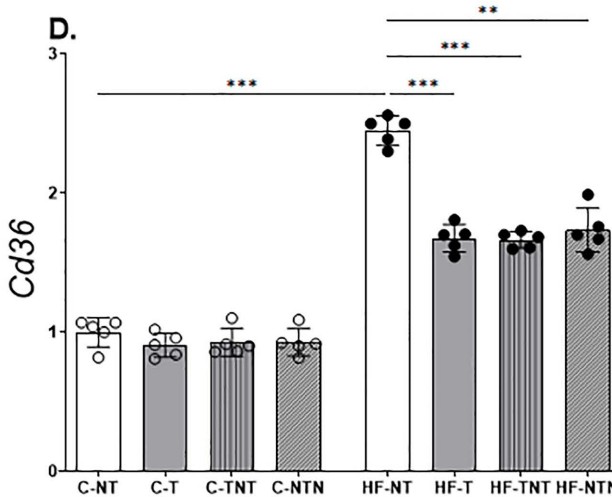

**Fig 4. Hepatic lipogenic pathway.** Gene expression was measured for sterol regulatory element-binding transcription factor 1 (Srebf1), MLX-interacting protein-like (Mlxipl), fas cell surface death receptor (Fas), and CD36 (Cd36). All gene expression levels were normalized to those of glyceraldehyde-3-phosphate dehydrogenase (Gapdh). Data are reported as the mean ± standard deviation, with significance at $p < 0.05$; $n = 5$. Statistical analysis was performed using one-way ANOVA with Brown–Forsythe and Welch correction. *$p < 0.05$, **$p < 0.01$, ***$p < 0.001$. Bar-ends indicate the pairwise group comparisons included in the statistical analysis.

The HF-NTN group, which underwent only late-cycle training following a period of detraining, showed a more limited reduction, with significant decreases in *Mlxpl* (−27%), *Fas* (−19%), and *Cd36* (−29%) levels, while *Srebf1* levels remained unaltered.

Both diet and HIIT significantly affected all lipogenic genes analyzed ($p < 0.0001$ for all), Fig 4.

**Hepatic lipogenic pathway (WB).** Compared with the C-NT group, the C-T (−62.5%), C-TNT (−73.8%), and C-NTN (−56.1%) groups exhibited significantly reduced hepatic SREBP-1c protein expression ($p < 0.01$). As expected, the HF-NT group showed a marked increase in SREBP-1c levels (+54.7%) relative to C-NT, confirming the lipogenic effect of the

high-fat diet. Relative to HF-NT, SREBP-1c abundance was significantly reduced in the HF-T (−36.8%), HF-TNT (−52.3%), and HF-NTN (−48%) groups (p < 0.01), indicating a consistent suppression of lipogenesis across both continuous and cyclic training protocols.

HIIT attenuated the diet-induced increase in CHREBP expression, in the control groups, training resulted in significant reductions in CHREBP levels in C-T (−78.0%), C-TNT (−56.3%), and C-NTN (−62.9%) (p < 0.05). Similarly, within the high-fat groups, HF-T exhibited a CHREBP expression lower than HF-NT (−78.0%) and showed additional significant reductions compared with HF-TNT (−97.6%) and HF-NTN (−95.0%) (p < 0.001).The two-way ANOVA confirmed that both diet and HIIT had significant effects on SREBP1c and CHREBP expression (Fig 7a and b, p < 0.0001).

**Hepatic β-oxidation pathway (RT-qPCR).** Compared with the C-NT group, the HF-NT group exhibited a marked downregulation of genes associated with hepatic fatty acid oxidation, with *Ppara* expression reduced by −61% and *Cpt1a* expression by −70% (Fig 5). Regarding the expression of *Ppara*, animals in the HF-NT group displayed markedly reduced mRNA levels compared with both trained high-fat groups (HF-T and HF-TNT; p = 0.004 and p = 0.03, respectively), indicating that HIIT was able to partially restore the transcriptional suppression induced by the high-fat diet. However, no significant difference was detected between HF-NT and HF-NTN (p = 0.98), suggesting that a single training cycle without continued exercise was insufficient to sustain transcriptional improvements. A comparable pattern was observed in the control diet groups, in which C-NT and C-NTN exhibited similar *Ppara* expression (p > 0.99), whereas training promoted detectable increases relative to sedentary controls.

Analysis of *Cpt1a* expression further supported these findings. In mice fed the control diet, both continuously trained animals and those trained in two cycles showed significantly higher *Cpt1-a* expression relative to C-NT, demonstrating that repeated exercise effectively stimulates hepatic fatty acid oxidation under basal dietary conditions. Consistently, within the high-fat groups, HF-T and HF-TNT exhibited significantly elevated *Cpt1-a* levels compared with HF-NT (p = 0.004 and p = 0.02, respectively). Nonetheless, *Cpt1-a* expression remained unchanged between HF-NT and HF-NTN (p = 0.95), reinforcing that detraining reduces the benefits induced by exercise when metabolic stress from the diet persists.

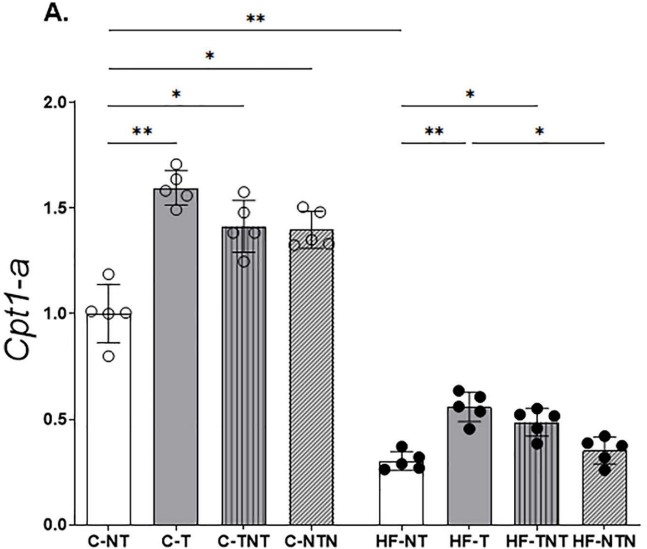
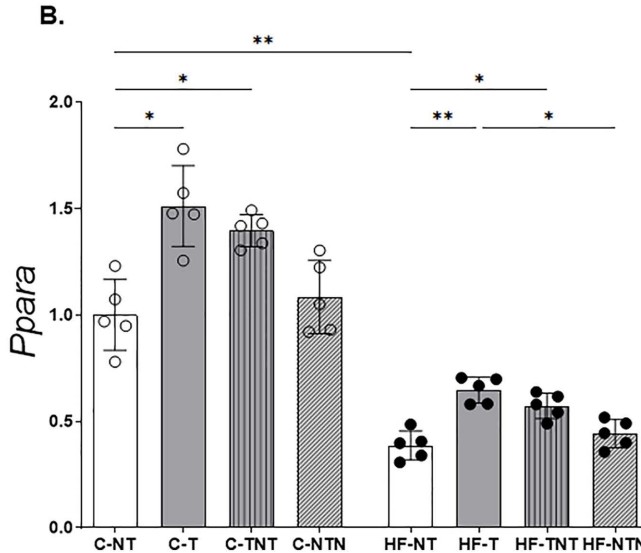

**Fig 5. Hepatic β-oxidation pathway.** The gene expression levels of carnitine palmitoyltransferase 1a (*Cpt1a*) and peroxisome proliferator activated receptor alpha (*Ppara*) were assessed and normalized to those of Gapdh. Data are reported as the mean ± standard deviation, with significance at p < 0.05; n = 5. Statistical analysis was performed using one-way ANOVA with Brown–Forsythe and Welch correction. *p < 0.05, **p < 0.01, ***p < 0.001. Bar-ends indicate the pairwise group comparisons included in the statistical analysis.

**Hepatic β-oxidation pathway (WB).** ACOX1 protein expression increased in response to HIIT, in control animals, the C-T group showed a significant elevation of +132.8% compared with C-NT (p<0.05). High-fat diet reduced ACOX1 levels in HF-NT (−11.5% vs. C-NT), consistent with diet-induced suppression of oxidative enzymes. Although ACOX1 levels were also higher in HF-T (+117.1%) and HF-TNT (+120.1%), these changes did not reach statistical significance. The two-way ANOVA showed a similar pattern for ACOX1, with both diet (p=0.01) and HIIT (p<0.0001) exerting significant effects, Fig 7c.

**Hepatic endoplasmic reticulum (ER) stress pathway (RT-qPCR).** Compared with those in the C-NT group, the expression levels of genes associated with ER stress, *Atf4* (+68%), *Ddit3* (+33%), and *Gadd45* (+84%), were significantly increased in the HF-NT group. In contrast, continuous HIIT in the HF-T group led to marked reductions in the expression of these genes, i.e., −33% (*Atf4*), −47% (*Ddit3*), and −33% (*Gadd45*) relative to those in the HF-NT group, Fig 6.

Similarly, the HF-TNT group, which underwent alternating training and detraining, showed comparable reductions: −32% (*Atf4*), −48% (*Ddit3*), and −28% (*Gadd45*). Compared with the HF-NT group, which was only trained during the final cycle, the HF-NTN group also exhibited decreased expression levels, albeit to a lesser extent (−21% *Atf4*, −24% *Ddit3*, and −17% *Gadd45*).

All groups that were fed a control diet presented significantly lower expression of *Atf4*, *Ddit3*, and *Gadd45*, reinforcing the protective effect of the control diet against diet-induced cellular stress. Both diet and HIIT significantly affected ER gene expression, with both factors showing strong effects (p<0.0001 for each gene), Fig 6.

**Hepatic ER stress pathway – GRP-78/BIP (WB).** Hepatic GRP-78/BIP protein expression was elevated in the HF-NT group (+108.7%; p<0.001) compared with C-NT. On the other hand, the HF-T exhibited a 62% reduction relative to HF-NT (p<0.001). Within the high-fat groups, HF-T also showed lower BIP/HSPA5 levels than both HF-TNT and HF-NTN (p<0.001 for both).

The two-way ANOVA confirmed that both diet (p<0.0001) and HIIT (p<0.0001) exerted significant effects on BIP expression, Fig 7d.

## Discussion

This study aimed to evaluate the effects of HIIT, including cycles of training, detraining, and retraining, on liver health and metabolic parameters in a mouse model involving a HF diet. Specifically, we investigated the response of metabolic disturbances induced by a HF diet to repeated cycles of exercise intervention.

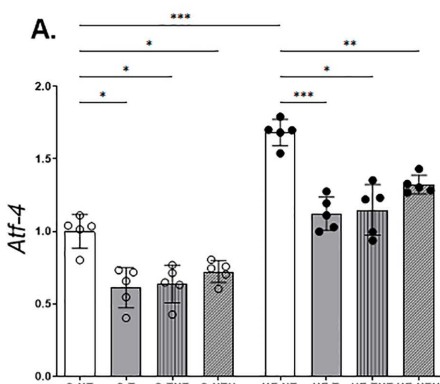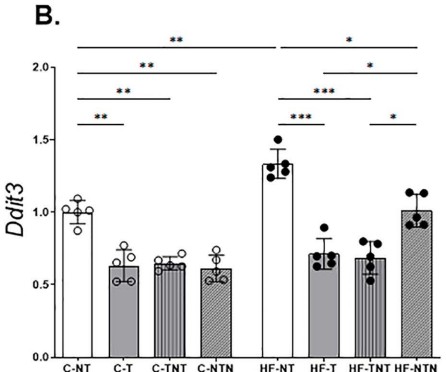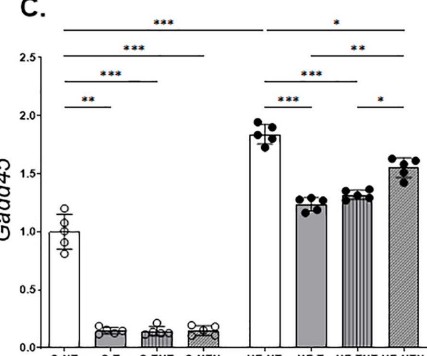

**Fig 6. Hepatic endoplasmic reticulum stress pathway.** Hepatic endoplasmic reticulum stress pathway. Gene expression was evaluated for transcription-activating factor 4 (Atf4), DNA damage-inducible transcript 3 (Ddit3), and growth arrest and DNA damage-inducible alpha (Gadd45), all normalized to Gapdh. Data are presented as mean±standard deviation (n=5). Statistical analysis was performed using one-way ANOVA with Brown–Forsythe and Welch corrections. *p<0.05, **p<0.01, ***p<0.001. Bar-ends indicate the pairwise group comparisons included in the statistical analysis.

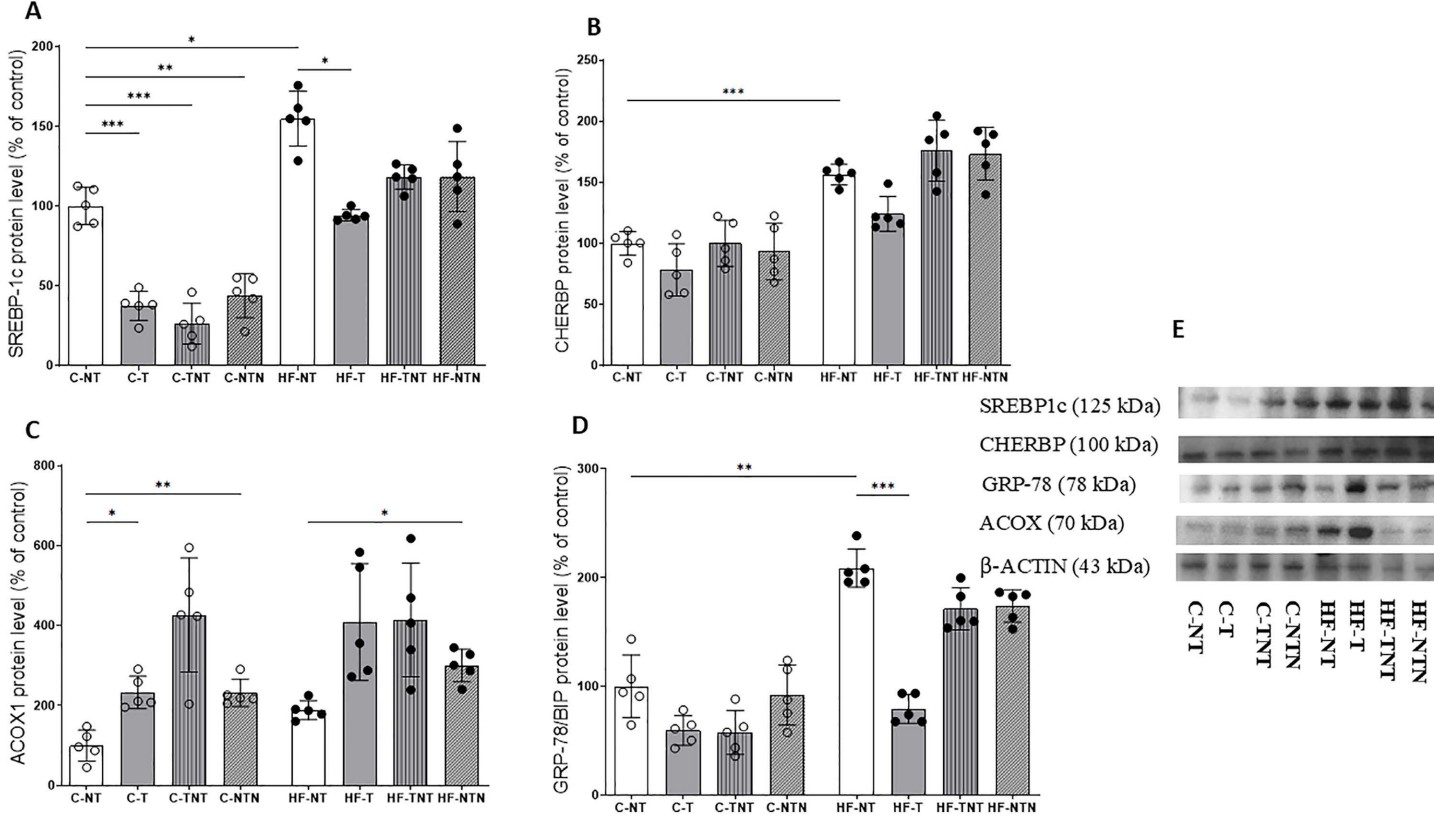

**Fig 7. Western blotting analysis of hepatic SREBP1c, CHREBP, ACOX1, BIP/HSPA5 protein expression.** The bands show the relative levels of these proteins across the experimental groups, normalized to that of beta actin as an internal control. All proteins were detected from the same membrane after sequential stripping and re-probing. The data are shown as the mean ± standard deviation, with significance at p < 0.05; n = 5. Statistical analysis was performed using one-way ANOVA with Brown–Forsythe and Welch correction. *p < 0.05, **p < 0.01, ***p < 0.001. Bar-ends indicate the pairwise group comparisons included in the statistical analysis.

Our findings revealed that compared with a control diet, an HF diet led to increased BM, insulin resistance (IR), glucose intolerance, and hepatic steatosis. In contrast, both continuous HIIT and training–detraining cycles produced beneficial metabolic effects, including reductions in BM, dyslipidemia, and insulin resistance. Importantly, HIIT consistently mitigated hepatic steatosis, and the extent of improvements correlated directly with the number of training cycles completed.

Although weight loss can be achieved through lifestyle interventions, including dietary modification and physical activity, maintaining weight loss long term remains a major challenge, with approximately 80% of individuals regaining the lost weight over time [21]. Sustained weight loss, however, is associated with a lower risk of cardiovascular disease, improved insulin sensitivity, and better outcomes in type 2 diabetes (T2D) prevention and management [22].

In the present study, after 12 weeks of either continuous training or alternating training and detraining cycles, compared with their sedentary counterpart groups, all the HIIT groups showed moderate reductions in BM. This finding is noteworthy, as HIIT is generally associated with modest weight loss [23]. Our data suggest that its metabolic benefits persist even during detraining periods, supporting its potential role in long-term weight management and liver health. Evidence from a recent systematic review of 24 exercise-only trials revealed that structured exercise (moderate to vigorous intensity, 3–5 days per week) can reduce hepatic steatosis by 20–30% [24,25].

These findings reinforce the consistent benefits of exercise on metabolic regulation, even in the absence of significant weight loss. In line with this evidence, less weight gain was observed in all HF diet-fed groups that underwent HIIT, whether for one, two, or three cycles, than in the sedentary HF diet-fed group. Moreover, the metabolic parameters in the groups that underwent training, detraining, and retraining cycles were comparable to those in the continuously trained group, suggesting that exercise cycling induced physiological adaptations that conferred a form of "metabolic memory" during detraining periods [5], as exemplified by the HF-TNT group.

To further elucidate these adaptations, the integration of FE, RQ, and EE data highlighted the metabolic modulation promoted by HIIT in animals fed a HF diet. The HF-NT group exhibited a greater FE and a lower RQ, indicating a predominance of lipid oxidation and greater energy storage. Although a reduction in FI (g/day) was observed in the HF diet-fed groups, the greater caloric density of the diet led to greater overall energy intake than in the control groups, which may explain the increased BM observed in these animals. In contrast, HIIT increased EE and reduced FE (HF-T and HF-TNT), suggesting greater caloric expenditure and improved substrate utilization. Notably, part of this effect persisted after the detraining period, reinforcing the lasting influence of exercise on energy metabolism and the attenuation of diet-induced metabolic alterations.

In the present study, both the continuously trained groups and those receiving training–detraining cycles showed a reduction in the area under the glucose curve over a 2-hour period, indicating improved glucose tolerance. These improvements are strongly associated with enhanced glucose uptake by skeletal muscle and increased insulin sensitivity [24]. Furthermore, it is well established that physical exercise reduces the area under the glucose curve in individuals with glucose intolerance [23].

Moreover, previous studies have demonstrated that HIIT can increase insulin sensitivity independent of weight loss or a reduction in adiposity, both in sedentary individuals and in experimental models of obesity or type 2 diabetes [24,26,27]. Our findings are consistent with these reports, reinforcing the potential of HIIT as an effective strategy to improve metabolic health, even in the absence of substantial changes in body composition.

In line with our findings, a study of 40 male Wistar rats that were fed an HF diet and subjected to 12 weeks of HIIT revealed improvements in the insulin response and in carbohydrate metabolism [28].

Regular physical training is widely recognized for its effectiveness in reducing body fat percentage, reducing serum triacylglycerol (TAG) and total cholesterol levels, and improving glucose homeostasis. However, normally discontinuing intensity-dependent training results in a decline in these benefits, with a regression of the physiological parameters that have been improved by exercise [29].

In a study in which female rats were fed an HF diet, HIIT inhibited the increase in subcutaneous and visceral adipose tissue and improved the lipid profile. This result was maintained after a 6-week detraining period, indicating the ability of this training to control adipocyte hypertrophy [30].

Although detraining is known to partially or completely reverse the physiological and metabolic adaptations induced by exercise [4], our study demonstrated that the training protocol was able to maintain improved insulin sensitivity in the groups that cycled through training, detraining, and retraining at levels comparable to those observed in the continuously trained group.

Regular physical training is highly effective for reducing the body fat percentage and lowering the serum levels of triacylglycerol (TAG) and total cholesterol while also contributing to the regulation of glucose homeostasis. However, the discontinuation of exercise often leads to the reversal of these physiological improvements [31].

In addition to enhancing insulin sensitivity, consistent physical activity promotes several metabolic benefits, including increased lipolysis, greater skeletal muscle mass, upregulation of β-oxidation pathways, improved intracellular lipid transport, and reduced oxidative stress [32]. In our study, mice fed a HF diet showed significant reductions in plasma total cholesterol and TAG levels following exercise interventions. This is an important outcome, given the role of these lipid markers in the progression and regression of liver disease [33].

The same behavior was observed in the analysis of ALT and AST, with the groups that consumed a lipid-rich diet and had greater exposure to training showing a reduction in these markers. This reduction is particularly interesting since high levels of these markers can impair liver function, cause cellular damage and lead to injury [34].

Additionally, we observed reduced expression of *Fas*, which is a rate-limiting enzyme in the terminal step of fatty acid biosynthesis [33]. This reduction likely contributed to the attenuation of hepatic steatosis observed in the exercised groups in this study.

With respect to the lipogenic pathway, all groups that underwent at least one training cycle presented a reduction in the expression of *Mlxipl*, a key transcription factor that, along with *Srebf1*, activates the enzymatic machinery responsible for converting excess glucose into fatty acids through the activity of *Fas* (fatty acid synthase) [35].

Given that HIIT relies heavily on glycometabolism, the significant downregulation of *Srebf1* and *Mlxipl* observed in trained HF diet-fed groups, even those subjected to detraining periods, may be attributed to elevated carbohydrate utilization during exercise, which suppresses glucose-induced lipogenesis [36].

Additionally, we observed reduced expression of *Fas*, a rate-limiting enzyme in the final step of fatty acid biosynthesis [37], which contributed to the improvement of hepatic steatosis observed in the present study.

Fatty acid metabolism is strongly regulated by the target genes of peroxisome proliferator-activated receptor-β/δ (*Ppar-β/δ*), which control the expression of key plasma membrane transporters such as *Cd36*. This transporter plays a central role in the uptake and translocation of long-chain fatty acids via facilitated diffusion [37]. In our study, HF diet-fed groups that underwent training across different cycles regardless of detraining exhibited reduced *Cd36* expression. This downregulation is likely associated with the lower hepatic TAG levels observed, suggesting a beneficial modulation of lipid uptake and storage by exercise. Consistent with the transcriptional reduction of key lipogenic regulators, the Western blot results confirmed a parallel decrease in SREBP-1c and CHREBP protein levels across all trained groups, including those subjected to detraining, demonstrating that HIIT effectively counteracted the diet-induced activation of hepatic lipogenesis at both gene and protein levels.

A study using C57BL/6 mice showed that compared with moderate-intensity exercise, HIIT is more effective for reducing the expression of genes involved in hepatic lipogenesis [38]. These findings suggest that the superior ability of HIIT to suppress fatty acid synthesis in the liver may be a key mechanism underlying its metabolic benefits [39].

In line with these results, our study demonstrates that the training cycle protocol significantly reduced hepatic fatty acid synthesis. Notably, this effect was also observed during detraining periods and became even more pronounced during retraining. This finding highlights the sustained and cumulative benefits of exercise cycling.

Based on these considerations, this study evaluated the expression of genes that play key roles in the hepatic ER stress pathway: *Atf4, Ddit3* (*Chop*), and *Gadd45*. *Atf4* is a transcription factor that regulates several genes associated with the unfolded protein response (UPR), including *Ddit3*, a major proapoptotic factor that mediates cellular responses to unresolved ER stress. *Gadd45*, in turn, functions as a downstream target of the CHOP–p38 signaling pathway and is typically induced during prolonged or severe ER stress, contributing to cell cycle arrest and apoptosis [40,41].

In our study, animals fed a HF diet presented elevated expression of *Atf4, Ddit3, and Gadd45*, indicating persistent ER stress and activation of apoptotic signaling. However, HIIT was effective at downregulating these markers in the groups that completed three training cycles. Remarkably, this reduction was also evident in groups that underwent detraining, suggesting a lasting protective effect of exercise against ER stress and its downstream apoptotic pathways. The protein data paralleled these gene expression changes, as GRP-78/BiP was similarly reduced in the trained groups, reinforcing the attenuation of hepatic ER stress. GRP-78/BiP acts as a key ER chaperone, maintaining protein-folding homeostasis and serving as a central regulator of the unfolded protein response during ER stress [42].

These findings align with those of Souza-Tavares et al. [43], who demonstrated that HIIT mitigates ER stress, enhances β-oxidation, preserves mitochondrial function, and attenuates hepatic steatosis. Taken together, our results indicate that physical training improves metabolic and cellular homeostasis in MASLD, even during periods of detraining, by reducing the expression of both early (Atf4/Ddit3) and late (Gadd45) markers of ER stress.

In conclusion, the animals retained, either partially or fully, the metabolic benefits acquired during the training periods, even throughout phases of detraining. These benefits, including improved glucose metabolism, lipid profiles, and body weight control, were more pronounced in groups that completed a greater number of training cycles. Notably, animals that underwent two training cycles exhibited more effective prevention of body weight regain, reinforcing the idea that repeated cycles of exercise increase the persistence of metabolic adaptations. These findings underscore the therapeutic value of continued or periodically repeated physical training in mitigating the adverse effects of a HF diet and preventing the progression of metabolic disorders such as MASLD.

## Supporting information

**S1 File. The images included in this file correspond to the original Western blot membranes used for protein expression analysis.**
(DOCX)

## Acknowledgments

The authors thank Aline Penna and Andrea Bertoldo for their technical assistance.

## Author contributions

**Conceptualization:** Guilherme Sá de Oliveira, Sandra Barbosa-da-Silva.

**Data curation:** Sandra Barbosa-da-Silva.

**Formal analysis:** Renata dos Santos Guarnieri, Guilherme Sá de Oliveira, Kaylaine Marques Ferreira, Aline Penna-de-Carvalho, Vanessa Souza-Mello, Sandra Barbosa-da-Silva.

**Funding acquisition:** Sandra Barbosa-da-Silva.

**Investigation:** Renata dos Santos Guarnieri, Aline Penna-de-Carvalho, Vanessa Souza-Mello, Sandra Barbosa-da-Silva.

**Methodology:** Guilherme Sá de Oliveira, Kaylaine Marques Ferreira, Aline Penna-de-Carvalho, Vanessa Souza-Mello, Sandra Barbosa-da-Silva.

**Project administration:** Renata dos Santos Guarnieri, Guilherme Sá de Oliveira, Sandra Barbosa-da-Silva.

**Resources:** Sandra Barbosa-da-Silva.

**Supervision:** Sandra Barbosa-da-Silva.

**Visualization:** Vanessa Souza-Mello.

**Writing – original draft:** Guilherme Sá de Oliveira, Sandra Barbosa-da-Silva.

**Writing – review & editing:** Renata dos Santos Guarnieri, Kaylaine Marques Ferreira, Aline Penna-de-Carvalho, Vanessa Souza-Mello, Sandra Barbosa-da-Silva.

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
