## [Decision Letter · Decision Letter 0]

5 Jan 2026

Dear Dr. Barbosa-Silva,

Thank you for submitting your manuscript to PLOS ONE. After careful consideration, we feel that it has merit but does not fully meet PLOS ONE’s publication criteria as it currently stands. Therefore, we invite you to submit a revised version of the manuscript that addresses the points raised during the review process.

We look forward to receiving your revised manuscript.

Kind regards,

Masoud Rahmati

Academic Editor

PLOS One

Journal Requirements:

3. Please include your tables as part of your main manuscript and remove the individual files. Please note that supplementary tables (should remain/ be uploaded) as separate "supporting information" files.

5. Please provide a complete Data Availability Statement in the submission form, ensuring you include all necessary access information or a reason for why you are unable to make your data freely accessible. If your research concerns only data provided within your submission, please write "All data are in the manuscript and/or supporting information files" as your Data Availability Statement.

Reviewer's Responses to Questions

**Comments to the Author**

1. Is the manuscript technically sound, and do the data support the conclusions?

Reviewer #1: Partly

2. Has the statistical analysis been performed appropriately and rigorously?

Reviewer #1: No

3. Have the authors made all data underlying the findings in their manuscript fully available?

Reviewer #1: No

4. Is the manuscript presented in an intelligible fashion and written in standard English?

Reviewer #1: Yes

Reviewer #1: This paper is a report of the effects of high intensity interval training on mice fed either control or high fat diets. The methods are described well, the statistics are mostly ok and just need more added to some figures, and the references are also good. The figures are clear. The overall methodology is that the authors put the mice through the diet and exercise program and then measure beta-oxidation and lipogenesis genes from liver tissue. I wonder if they could have done some of these analyses in parallel with adipose tissue which would have been informative.

Line 328. Please do not use abbreviations in subtitles. FI and EE are defined in the text but IC and FE are not within this section. I found them in prior paragraphs, but it just makes reading more cumbersome.

Diet information (table 1) is missing. In fact, all the tables are missing! This is why I had to mark "no" on all data fully available even though most of it is described in the text of the paper.

Fig. 2 needs correction in that a ≠ C-NT since there is no plain "C". Please include any statistical data in the figure legend.

Fig. 3 Data should be quantified by assessing triglycerides in liver tissue.

Fig. 4 A statement in the figure legend should say something similar to "the bar-ends signify which two groups are represented for statistical analysis"

Fig. 5 were HF-NT and HF-NTN compared and C-NT and C-NTN? Were they not significant?

Fig. 7 More details are needed in the methods section for the WB data. These all should have come from the same blot so what were the stripping methods used when assessing different protein levels?

**Do you want your identity to be public for this peer review?** For information about this choice, including consent withdrawal, please see our Privacy Policy

Reviewer #1: No

---

## [Author Response · Author response to Decision Letter 1]

21 Jan 2026

PONE-D-25-65118

Persistence of the hepatic benefits of high-intensity interval training (HIIT) during detraining despite body weight regain in mice.

PLOS One

Dear Masoud Rahmati

I am pleased to have the opportunity to address the questions raised by the reviewers, as well as the corrections suggested by the Academic Editor. We trust that our responses meet the requests, and to facilitate evaluation and clarity, all revisions have been highlighted in the updated version of the Revised Manuscript with Track Changes.

Kind regards,

Sandra Barbosa-da-Silva

A letter that responds to each point raised by the academic editor and reviewer(s). You should upload this letter as a separate file labeled 'Response to Reviewers'.

Response: A version was added.

Na unmarked version of your revised paper without tracked changes. You should upload this as a separate file labeled ‘Manuscript’.

Response: A version was added.

We look forward to receiving your revised manuscript.

Kind regards,

Masoud Rahmati

Academic Editor, PLOS One

Journal Requirements:

Response: All blots were added to a file as supplementary material.

3. Please include your tables as part of your main manuscript and remove the individual files. Please note that supplementary tables (should remain/ be uploaded) as separate "supporting information" files.

Response: All tables were included in the main article.

Response: Grant information was corrected: RSG and GSO received a bursary from the Coordenação de Aperfeiçoamento de Pessoal de Nível Superior (CAPES, Brazil), Finance Code 001. The study was supported by Fundação Carlos Chagas Filho de Amparo à Pesquisa do Estado do Rio de Janeiro (FAPERJ), grant number: 26/210.743/2024. The funders had no role in study design, analysis, decision to publish,or preparation of the manuscript

5. Please provide a complete Data Availability Statement in the submission form, ensuring you include all necessary access information or a reason for why you are unable to make your data freely accessible. If your research concerns only data provided within your submission, please write "All data are in the manuscript and/or supporting information files" as your Data Availability Statement.

Response: The statement was added with the suggested phrase: “All data are in the manuscript and/or supporting information files”

Response: No specific request for additional citations was made by the reviewers.

Review Comments to the Author

Reviewer #1: This paper is a report of the effects of high intensity interval training on mice fed either control or high fat diets. The methods are described well, the statistics are mostly ok and just need more added to some figures, and the references are also good. The figures are clear. The overall methodology is that the authors put the mice through the diet and exercise program and then measure beta-oxidation and lipogenesis genes from liver tissue. I wonder if they could have done some of these analyses in parallel with adipose tissue which would have been informative.

Response: We thank the reviewer for the insightful comment and the suggestion to perform parallel analyses in adipose tissue. The first hepatic findings prompted us to also explore exercise-induced adaptations in adipose tissue, particularly regarding metabolic and inflammatory pathways. However, we did not harvest the visceral adipose tissue in this experiment. Surely, in our next experiment, we will dive into the adipose tissue and explore its crosstalk with the liver.

2 - Line 328. Please do not use abbreviations in subtitles. FI and EE are defined in the text but IC and FE are not within this section. I found them in prior paragraphs, but it just makes reading more cumbersome.

Response: Thank the reviewer for pointing out the issue regarding the use of abbreviations in subtitles. As suggested, we have removed the abbreviations and replaced them with the full terms in the title section. Specifically, “FI,” “IC,” “EE,” and “FE” were revised to “Food Intake,” “Indirect Calorimetry,” “Energy Expenditure,” and “Feed Efficiency,” respectively.

We appreciate the reviewer’s careful attention and constructive recommendations.

3- Diet information (table 1) is missing. In fact, all the tables are missing! This is why I had to mark "no" on all data fully available even though most of it is described in the text of the paper.

Response: Thank the reviewer for bringing this issue to our attention and sincerely apologize for the inconvenience.

The tables, including the diet composition (Table 1), were originally prepared and submitted. However, it appears that due to a processing or formatting issue during submission, they were not properly displayed in the uploaded version, resulting in their omission from the file viewed by the reviewers.

Following the editor’s recommendation, we have now inserted all tables directly into the main manuscript text to ensure full visibility and accessibility. This includes Table 1 with detailed diet information, as well as the remaining tables referenced throughout the paper.

We appreciate the reviewer’s careful reading and patience, and we apologize again for the oversight.

4-Fig. 2 needs correction in that a ≠ C-NT since there is no plain "C". Please include any statistical data in the figure legend.

Response: We sincerely apologize for the mistake in Figure 2. The labeling error has now been corrected to properly reflect the comparison with the C-NT group, and the figure has been updated accordingly.

In addition, we have revised the figure legend to include the corresponding statistical information, as requested.

We appreciate the reviewer’s attention to detail and thank you for helping us improve the clarity and accuracy of the manuscript.

Legend: Fig. 2. Body mass evolution. Data are presented as the mean and standard deviation, with significance determined at p < 0.05; n=5 per group. The following symbols are used: [a] ≠ C-NT and [b] ≠ HF-NT. In the pre-HIIT weeks, Student’s t test with Welch correction was performed for comparisons. Groups: C-NT, untrained control; C-T, control trained during the 3 cycles; C-TNT, control trained in cycle 1, not trained in cycle 2 and trained in cycle 3; C-NTN, control not trained in cycle 1, trained in cycle 2 and not trained in cycle 3; HF-NT, high-fat diet not trained; HF-T, high-fat diet trained during the 3 cycles; HF-TNT, high-fat diet trained in cycle 1, not trained in cycle 2 and trained in cycle 3; HF-NTN, high-fat diet not trained in cycle 1, trained in cycle 2 and not trained in cycle 3

5-Fig. 3 Data should be quantified by assessing triglycerides in liver tissue.

Response: Thank you for your valuable comments and the opportunity to clarify our methodological approach. The photomicrographs in Figure 3 illustrate the assessment of hepatic steatosis using stereological quantification of volume density. This morphometric analysis enabled us to estimate the relative abundance of lipid droplets within hepatocytes as a structural indicator of liver lipid accumulation.

In addition, hepatic triglyceride content was determined using a biochemical method based on spectrophotometric detection from liver tissue extracts. These quantitative biochemical data are reported in Table 5 of the manuscript.

6- Fig. 4 A statement in the figure legend should say something similar to "the bar-ends signify which two groups are represented for statistical analysis"

Response: Revised figure legend

Hepatic endoplasmic reticulum stress pathway. Gene expression was evaluated for transcription-activating factor 4 (Atf4), DNA damage-inducible transcript 3 (Ddit3), and growth arrest and DNA damage-inducible alpha (Gadd45), all normalized to Gapdh. Data are presented as mean ± standard deviation (n = 5). Statistical analysis was performed using one-way ANOVA with Brown–Forsythe and Welch corrections. *p < 0.05, **p < 0.01, ***p < 0.001. Bar-ends indicate the pairwise group comparisons included in the statistical analysis.

7-Fig. 5 were HF-NT and HF-NTN compared and C-NT and C-NTN? Were they not significant?

Response: Thank you for this observation. We confirm that the pairwise comparisons between HF-NT vs. HF-NTN and C-NT vs. C-NTN were performed as requested. As shown in Figure 5, no significant differences were detected between HF-NT and HF-NTN for either Ppara (p = 0.98) or Cpt1a (p = 0.95), indicating that a single HIIT cycle without sustained training was insufficient to modify the expression of β-oxidation genes in the high-fat diet context.

Similarly, C-NT and C-NTN did not differ significantly for Ppara expression (p > 0.99), suggesting that detraining did not alter basal transcriptional levels in animals receiving the control diet. These data have now been explicitly described in the Results section, page 19, lines 560-580.

8- Fig. 7 More details are needed in the methods section for the WB data. These all should have come from the same blot so what were the stripping methods used when assessing different protein levels?

Response: We appreciate the reviewer's insightful feedback. The Methods section now contains more methodological information (page 10, lines 327–337). The same membrane displayed all the proteins in Fig. 7. Following the manufacturer's instructions, membranes were stripped with Restore TM PLUS Western Blot Stripping Buffer (Millipore Sigma) after each target protein was identified. The membranes were then re-probed using the relevant primary antibodies. Additionally, the figure 7 legend (page 30, line 977) now includes this information. The loading control for normalization was beta-actin.

Methods:

Protein levels related to lipogenesis, beta-oxidation, and endoplasmic reticulum stress were analyzed by Western blotting. Details of the primary antibodies are provided in Table 3. All target proteins were detected from the same membrane. After chemiluminescent detection of each protein, membranes were stripped using Restore™ PLUS Western Blot Stripping Buffer (Millipore Sigma), following the manufacturer’s instructions, and subsequently re-probed with the corresponding primary antibodies. β-Actin was used as a loading control. Protein detection was performed by electrochemiluminescence using the ChemiDoc XRS Molecular Imaging System (Bio-Rad, Hercules, CA, USA). Band intensities (chemiluminescence signals) were quantified using ImageJ software (version 1.54g; NIH, USA).

Figure 7 legend: Western blotting analysis of hepatic SREBP1c, CHREBP, ACOX1, and BIP/HSPA5 protein expression. The bands show the relative levels of these proteins across the experimental groups, normalized to beta-actin as an internal control. All proteins were detected from the same membrane after sequential stripping and re-probing. The data are shown as the mean ± standard deviation, with significance at p < 0.05; n=5. Statistical analysis was performed using one-way ANOVA with Brown–Forsythe and Welch correction. *p < 0.05, **p < 0.01, ***p < 0.001.

---

## [Decision Letter · Decision Letter 1]

28 Jan 2026

Persistence of the hepatic benefits of high-intensity interval training (HIIT) during detraining despite body weight regain in mice.

PONE-D-25-65118R1

Dear Dr. Barbosa-Silva,

We’re pleased to inform you that your manuscript has been judged scientifically suitable for publication and will be formally accepted for publication once it meets all outstanding technical requirements.

Kind regards,

Masoud Rahmati

Academic Editor

PLOS One

Additional Editor Comments (optional):

Reviewers' comments:

Reviewer's Responses to Questions

**Comments to the Author**

Reviewer #1: All comments have been addressed

2. Is the manuscript technically sound, and do the data support the conclusions?

Reviewer #1: Yes

3. Has the statistical analysis been performed appropriately and rigorously?

Reviewer #1: Yes

4. Have the authors made all data underlying the findings in their manuscript fully available?

Reviewer #1: Yes

5. Is the manuscript presented in an intelligible fashion and written in standard English?

Reviewer #1: Yes

Reviewer #1: All concerns have been addressed in this revised manuscript. I think it will have an interest with a broad audience, especially those who pay gym memberships to do these HIIT programs.

**Do you want your identity to be public for this peer review?** For information about this choice, including consent withdrawal, please see our Privacy Policy

Reviewer #1: **Yes:** Edward N Harris

---

## [Editor Report · Acceptance letter]

PONE-D-25-65118R1

PLOS One

Dear Dr. Barbosa-da-Silva,

I'm pleased to inform you that your manuscript has been deemed suitable for publication in PLOS One. Congratulations! Your manuscript is now being handed over to our production team.

Kind regards,

on behalf of

Dr. Masoud Rahmati

Academic Editor

PLOS One